# Successful Treatment of Cryptococcal Meningitis and Cryptococcoma with Isavuconazole in a Patient Living with HIV

**DOI:** 10.3390/jof7060425

**Published:** 2021-05-28

**Authors:** Brendan O’Kelly, Aia Mohamed, Colm Bergin, Fiona Lyons, Thomas R. Rogers, Brian O’Connell, Emma Devitt

**Affiliations:** 1Department of Genitourinary Medicine and Infectious Diseases (GUIDe), St James’s Hospital, Dublin 8, Ireland; aiamohamed@stjames.ie (A.M.); cbergin@stjames.ie (C.B.); Flyons@stjames.ie (F.L.); emdevitt@stjames.ie (E.D.); 2Department of Clinical Microbiology St James’s Hospital and Trinity College Dublin, Dublin 8, Ireland; trogers@stjames.ie (T.R.R.); BOConnell@stjames.ie (B.O.)

**Keywords:** cryptococcal meningitis, HIV, IRIS, fluconazole resistance, AUC:MIC ratio, isavuconazole

## Abstract

We describe the successful use of isavuconazole for treatment of an HIV-positive patient with cryptococcal meningitis following induction therapy with liposomal amphotericin B and flucytosine. Because the Cryptococcus neoformans isolate from cerebrospinal fluid had a borderline minimum inhibitory concentration of 8 mg/L, initial consolidation therapy was given with a daily dose of fluconazole 1200 mg based on area under the curve to minimum inhibitory concentration modelling data. Toxicity, and the radiological emergence of a cryptococcoma in the setting of immune reconstitution inflammatory syndrome, prompted a therapeutic switch to isavuconazole. Subsequent imaging after 19 weeks of isavuconazole shows a significant reduction in cryptococcoma size from 11 mm to complete resolution. The patient remains well after 210 days of therapy with a view to completion of treatment after 1 year.

## 1. Introduction

Cryptococcal meningitis (CM) traditionally has a poor prognosis and most commonly occurs in patients with advanced AIDS [1]. Most consensus treatment guidelines recommend a combination of Liposomal amphotericin B (Ambisome™) plus flucytosine for 2 weeks of induction therapy [2,3,4]. Fluconazole is the mainstay drug for the subsequent 8-week continuation phase at doses of 400 mg or higher, then for the subsequent maintenance phase at a dose of 200 mg for up to 6 months or longer if HIV is not suppressed or CD4 count is less than 100 cells/mm^3^. Fluconazole has assumed this role because of its therapeutic efficacy, oral route of administration, and safety profile over prolonged periods. 

Despite the availability of other triazoles like itraconazole, voriconazole, posaconaozle, and most recently, isavuconazole, they have yet to find an established role in the management of CM. Itraconazole has been shown to be inferior to fluconazole in the management of CM and is recommended only as a second-line therapy where fluconazole is either too toxic or the cryptococcal isolate has reduced susceptibility to fluconazole [2]. Similarly, voriconazole has not established itself in CM management despite having excellent CNS penetration and being the antifungal drug of choice for invasive *Aspergillus* spp. CNS infections [5]. Access to voriconazole in countries with higher case numbers and mitigation of its associated toxicities could be difficult in resource limited settings (RLS). Isavuconazole, the most recently licensed triazole, is less well studied in CM. It has reduced toxicity compared to voriconazole, and does achieve therapeutic levels in cerebrospinal fluid (CSF) in those with CNS infection [6] but is more expensive than fluconazole. This drug shows promise for the management of CM, yet to date, is understudied for this role [7,8]. 

## 2. Case

A 38-year-old Caucasian man from sub-Saharan Africa presented to the emergency department with a 2-week history of progressively worsening occipital headache which was worst on waking. He also had generalised malaise and weight loss of 6 kg in the preceding three months. He described blurred vision when headaches were particularly severe, and no other focal neurological deficits. There were no symptoms of fever, chills, shortness of breath, cough, odynophagia, diarrhoea, or skin changes to suggest other opportunistic infections.

His background history includes HIV which was diagnosed in 2003 when he presented with presyncope and tachyarrhythmia to another hospital and was found to have a 1 litre pericardial tamponade caused by *Mycobacterium tuberculosis*. The organism was found to be pan-sensitive and the patient was commenced on rifampicin/isoniazid/pyrazinamide/ethambutol and steroid induction followed by rifampicin and isoniazid to complete ten months of therapy. CD4 nadir was 100 cell/mm^3^. The patient was commenced on zidovudine/lamivudine and efavirenz combination ART at that time. In 2007, treatment was rationalised to single tablet regimen; tenofovir disoproxil fumarate/emtricitabine/efavirenz. The patient was fully adherent, achieved HIV viral suppression, and had no additional infections or complications of treatment until his loss to follow up in 2010. CD4 count recovered to 697 (32%) cells/mm^3^ at that time. The patient did not seek care at another institution and remained off ART for the following ten years. 

On admission to our institution, the patient’s Glasgow Coma Score was 15/15. The patient had intact neurological, cardiovascular, and respiratory systems on examination. The patient complained of an occipital headache and pain was elicited on neck flexion although no overt stiffness was seen. Brudzinski sign was positive. No skin rashes, palatal, or genital rashes were found.

A serum cryptococcal antigen (CRAG) (CrAG ^®^ LFA Cryptococcal lateral flow Assay (IMMY, Inc, Norman, Oklahoma, USA) was positive 1:1280. A lumbar puncture (LP) was performed after a contrast magnetic resonance image (MRI) of the brain ruled out space occupying lesion or midline shift. An opening pressure (OP) of 25 cm H_2_0 was found, as well as a CRAG 1:160, a peak white cell count of 127 cells/mm^3^ (range 0–5 cells/mm^3^, 95% mononuclear), protein 91 mg/dl (CSF range: 15–45 mg/dL), and a CSF glucose 2.19 mmol/L (range 2.22–2.39 mmol/L) with a paired serum glucose of 4.8 mmol/L (range 3.5–7.7 mmol/L). No organisms were seen on CSF microscopy but *Cryptococcus neoformans* was grown on day 5 on Sabouraud dextrose agar medium. Other laboratory findings include: CD4 count 43 (9%) cells/mm^3^ (range 502–1749 cells/mm^3^), HIV viral load 148,402 (log 5.17) copies/mL, haemoglobin 10.4 g/dL (range 13.5–18 g/dL), lymphocyte count 0.4 × 10^9^/L (range 1.5–3.5 × 10^9^/L), platelet count 160 × 10^9^/L (range 140–450 × 10^9^/L), neutrophils 1.2 × 10^9^/L (range 2.0–7.5 × 10^9^/L), serum beta-d-glucan 91 pg/mL (range < 8 to 60 pg/mL), and Anti-Hepatitis B surface antibody 54 IU/mL. A chest radiograph showed clear lung fields. 

Induction therapy with Liposomal Amphoterocin-B (Ambisome™) 4 mg/kg 24 hourly and flucytosine 25 mg/kg 6 hourly was commenced. Daily LPs were done initially until OP was consistently below 15 cm H_2_0. Flucytosine was discontinued on day 12 due to thrombocytopenia. On day 14, repeat CSF culture was sent, Ambisome™ was continued until cultures were confirmed negative on day 21. Susceptibility testing indicated that the fluconazole MIC was 8.0 mg/L; 0.125 mg/L for isavuconazole; voriconazole 0.25 mg/L; itraconazole 0.12 mg/L; amphotericin B 0.25 mg/L and flucytosine 2 mg/L on the Clinical and Laboratory Standards Institute (CLSI) [9]. Fluconazole 1200 mg daily was used for the consolidation phase of treatment, using a fluconazole exposure prediction model based on the area under the curve to minimum inhibitory concentration (AUC:MIC) ratio [10]. The patient was discharged at the end of intravenous induction. A timeline of patient progress and therapy can be seen in Figure 1. Tenofovir alafenamide/emtricitabine/bictegravir triple therapy was commenced on day 26. By day 34, the patient was re-admitted with intractable nausea and vomiting, acute kidney injury, alopecia, and mucositis. Fluconazole 1200 mg was continued initially, then reduced to 800 mg OD on day 45. ART was amended to emtricitabine, rilpivirine/dolutegravir to avoid tenofovir in light of renal impairment. Another brain MRI with contrast was done on day 67 to identify a central cause of nausea; two new lesions were identified on fluid-attenuation inversion recovery (FLAIR) MRI sequence in the right globus pallidus and left frontal lobe consistent with cryptococcomas; Figure 2. Fluconazole was stopped and the patient received another 2 weeks of Ambisome™ induction of 4 mg/kg then switched to isavuconazole 200 mg twice daily loading followed by 200 mg once daily thereafter. Prednisolone 60 mg daily was also commenced on the discovery of the cryptococcomas and was tapered in the subsequent weeks. The cryptococcoma of the right global pallidus has reduced in size from 11 mm to 4 mm by day 94 of treatment and completely resolved by day 210. The second lesion of the left frontal lobe in retrospect was reported to be a meningioma. The cryptococcoma is thought to have been present sub-clinically and to have emerged as a phenomenon of immune reconstitution inflammatory syndrome in the context of recently commenced HIV therapy rather than as a result of treatment failure with fluconazole.

Unfortunately, the patient missed doses of both ART and isavuconazole in January 2021 whilst at home due to COVID-19 lockdown isolation, which was reflected in bloods on day 119 post initial diagnosis; HIV VL 11,875 copies/mL and isavuconazole level 1.68 mg/L. The patient was reloaded with isavuconazole and recommenced ART with close follow up and therapeutic drug monitoring. A new M184V mutation was found on HIV genotypic resistance testing at that time. The patient is currently in week 15 of therapy with isavuconazole and doing well with a view to stopping therapy per guidelines [2].

## 3. Discussion

The established treatment of CM is the combination therapy of amphotericin B deoxycholate and flucytosine. This combination has been shown to be the most fungicidal, to have the greatest efficacy in sterilising CSF cultures by the end of the induction phase, and to be associated with reduced mortality [11]. International guidelines including those from the Infectious Diseases Society of America (IDSA), the Centers for Disease Control/National Institute of Health (CDC/NIH), and the British HIV Association (BHIVA), recommend two weeks of induction therapy with this combination [2,3,4]. It does appear that in RLS, there is an increased risk of mortality (38% (95% CI: 29–32%) vs. 24% (95% CI: 16–32%)) when using these drugs for 2 weeks versus 1 week [12]. This may be due to difficulties with early identification of adverse events like acute kidney injury (AKI), hypokalaemia, liver injury, and cytopenias in an RLS and mitigation of these complications. For this reason, the WHO recommends one week of Amphotericin B plus flucytosine, then one week of 1200 mg fluconazole for the 2 week induction of CM treatment [13]. Notably, of the 10 countries with the highest incidence of CM, at least 8 do not have access to flucytosine [14]. Despite international guideline recommendations, the vast majority of cases of CM are managed with high dose fluconazole for reasons outlined above; access to medication, mitigating of adverse events due to amphotericin B and flucytosine toxicity, and previous experience with fluconazole.

CSLI epidemiological cut-off values are the current standard in describing susceptibility of cryptococcus spp. to antifungal therapy [9]. At present, it is not certain that decreased susceptibility to fluconazole is associated with worse outcomes. One small study of 35 patients with cryptococcal infections treated with fluconazole, 13 (37.1%) of whom had with an MIC ≥ 16 mg/L, showed no significant difference in mortality between those with MICs ≥ 16 mg/L or below [15,16]. A systematic review of fluconazole resistance in clinical isolates of *Cryptococcus* species found that isolates with in vitro resistance were more likely to be associated with clinical relapse [17]. Certainly, it appears that decreasing azole susceptibility is being recognised among isolates of *C. neoformans*. The ARTEMIS DISK Antifungal Surveillance Study 1997–2007, which included over 3600 cryptococcal isolates, found resistance in 7.3% of isolates between 1997–2000 and 11.7% resistant from 2005–2007 [18] using CLSI disk diffusion standard M44-A. One way forward is to use the AUC:MIC ratio as a tool to guide therapeutic fluconazole dosing. As fluconazole follows first order kinetics, it has been inferred that a ratio >389 is key to treating CM and that dose adjustment can be done in a meaningful way to achieve a cure in patients with resistant organisms [10]. The offset to this approach is the risk of toxicity due to increased drug exposure.

In this case, the patient was treated with this approach, using a higher fluconazole dose of 1200 mg daily based on the AUC:MIC ratio modelling data [10]. The increased toxicity of using high-dose fluconazole was also seen in this case and led to re-admission. Fluconazole toxicity has been reported at a rate as high as 51.6% in patients receiving long-term treatment for coccidioidomycosis in one study. Of the 64 patients who developed toxicity, the median dose was 6.7 mg/kg daily (most commonly 400 mg daily, (range 100–1600 mg daily)) and median time to adverse effects was 119 days. Interestingly, alopecia, which was seen in our patient, occurred in 16.1% of patients in this study [19]. Increased doses of fluconazole may also lead to increased risk of drug–drug interactions.

In light of rising rates of fluconazole resistance, the need for higher dosing of fluconazole using strategies like AUC:MIC based dosing is becoming increasingly important. Inevitably, this will translate to higher toxicity and the need for alternatives to fluconazole for CM management. These alternatives will need to have acceptable safety and tolerability profiles that patients can take relatively unmonitored for weeks at a time and tolerate for months. Isavuconazole may be a suitable candidate in this role. In a mouse model, it has been shown to be efficacious in the management of CM [20]. Reports of its use in human cases are scant. One report of isavuconazole treatment in 36 fungal CNS infections describe five cases of CM (three due to *C. neoformans,* two due to *C. gattii*) all of whom were alive by day 84 [7]. It is unclear if, in these cases, isavuconazole was used as first-line or salvage therapy. In the VITAL study, an open-label phase III trial exploring isavuconazole for rare fungal pathogens, nine patients had cryptococcal disease either alone or in combination with other opportunistic infections like nocardiosis. CM was seen in six patients, three of whom received isavuconazole as primary therapy and three as salvage therapy. The median duration of therapy was 180 days, and all patients with CM survived [8].

Voriconazole has been shown to have efficacy in CM. In a study of 55 patients who received either voriconazole (*n* = 16), fluconazole plus amphotericin B (*n* = 28), or flucytosine plus amphotericin B (*n* = 11) in a centre in China [21], the best outcomes were seen in the voriconazole group with a 100% response rate compared to 67.9% and 27.3% seen in the other two groups, respectively [21]. Despite these results with voriconazole, its numerous toxicities and the challenge of many drug:drug interactions put it at a disadvantage when compared to isavuconazole as a therapeutic regimen that may be needed for several months’ duration as is the case with CM.

At present, isavuconazole is expensive and inaccessible in places where most CM cases are found. With time, drugs become generically available and less expensive—this speaks to the question of future “suitability” of isavuconazole use in RLS. The need for TDM is a challenge at present and may be the biggest barrier to its use in RLS.

We have shown in this case that isavuconazole can be used successfully in a resource-abundant setting for the continuation and maintenance phases of CM management. Randomised trials are needed to explore isavuconazole’s efficacy for CM.

## Figures and Tables

**Figure 1 jof-07-00425-f001:**
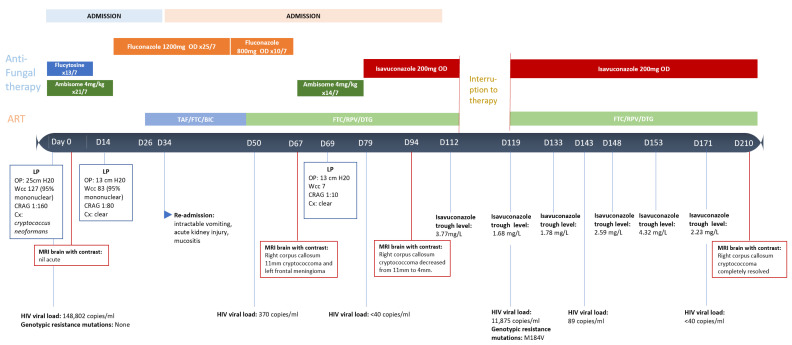
Timeline of patient progression.

**Figure 2 jof-07-00425-f002:**
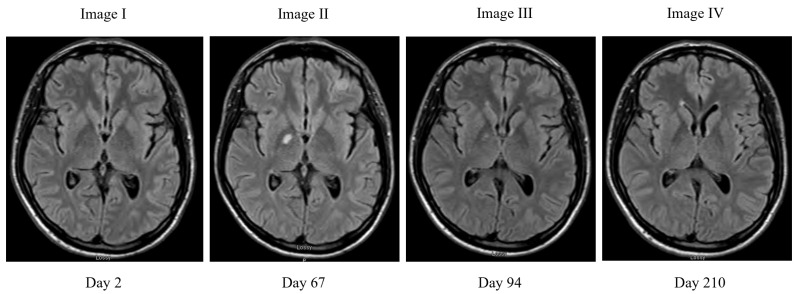
MRI FLAIR changes over time. Image I on admission shows no masses, Image II shows 11 mm enhancing masse in right corpus callosum and left frontal meningioma, Image III shows improvement in the right corpus callosum mass to 4 mm, Image IV shows complete resolution of the mass.

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
