# Peer review of "Successful Treatment of Cryptococcal Meningitis and Cryptococcoma with Isavuconazole in a Patient Living with HIV"

_jof, 2021, doi:10.3390/jof7060425_

Round 1
Reviewer 1 Report
To many abreviations. Some of them are not used more than once. e.g. CRAG, ED, ART, LP, RLS, OP, TB, GCS, IRIS, HBs, etc. That's make reading difficult. Other abreviations were correctly used but they were defined after authors used a couple of times before (CSF was used in lines 76 and 77 and defined in line 79).
Abstract: Please avoid abreviations in the abstract.
line 34: is ".... only as a second -ine therapy" "...only as a second-line therapy"?
line 38: spp. is missing or or there is a point left.
Lines 38-46: It should be rewritten. Moreover, I do not think that a decrease in the number of cryptoccocus infections is a reason for not using voriconazole. Prove me wrong with a reference.
Line 83: Beta-d-glucan quantification was performed in serum or in CSF?
Line 84: is Anti-HBs antibodies against Hepatitis B? Please define.
Line 89: How was the susceptibility testing performed? The used protocol/method should be defined since epidemiological cut off values varies depending on method (and in genetic variety of the studied Cryptococcus).
Line 100: Define MRI. It should not be better "brain MRI?"
Line 101: What is FLAIR?
Line 107: a space should be deleted between the two numbers 1 (11 mm. not 1 1 mm.).
Line 139: The mentioned breakpoints are completly obsolete. There is no mention of the used protocol (e.g. CLSI or EUCAST to obtain the isolate MIC). Each protocol has their own cut off values. The active CLSI protocols (M27 4th ed. M59 rd ed.) have epidemiological cut off values or ECVs (not breakpoints) for Cryptococcus neoformans VNI. For this genetic variant, fluconazole ECV is 8ug/ml. Thus, if the strain was not genotiped, and no susceptibility testing used method is mentioned no interpretation is posible.
Although established breakpoints for fluconazole in CM are MIC ≤8.0 mg/L suscep- 139
tible, 16-32 μg/mL dose-dependent susceptible and ≥64 μg/mL resistant, it is unclear 140
whether significantly adverse outcomes are seen in those with decreased susceptibility 14.
Author Response
Dear reviewer thank you for your comments, the manuscript has been substantially improved with these amendments
To many abreviations. Some of them are not used more than once. e.g. CRAG, ED, ART, LP, RLS, OP, TB, GCS, IRIS, HBs, etc. That's make reading difficult. Other abreviations were correctly used but they were defined after authors used a couple of times before (CSF was used in lines 76 and 77 and defined in line 79). Amended
Abstract: Please avoid abreviations in the abstract. Amended
line 34: is ".... only as a second -ine therapy" "...only as a second-line therapy"? Amended
line 38: spp. is missing or or there is a point left. Amended
Lines 38-46: It should be rewritten. Moreover, I do not think that a decrease in the number of cryptoccocus infections is a reason for not using voriconazole. Prove me wrong with a reference. Amended, sentence restructured
Line 83: Beta-d-glucan quantification was performed in serum or in CSF? Amended - serum
Line 84: is Anti-HBs antibodies against Hepatitis B? Please define. Amended
Line 89: How was the susceptibility testing performed? The used protocol/method should be defined since epidemiological cut off values varies depending on method (and in genetic variety of the studied Cryptococcus). Amended, CLSI
Line 100: Define MRI. It should not be better "brain MRI?" Amended
Line 101: What is FLAIR? Amended
Line 107: a space should be deleted between the two numbers 1 (11 mm. not 1 1 mm.). Amended
Line 139: The mentioned breakpoints are completly obsolete. There is no mention of the used protocol (e.g. CLSI or EUCAST to obtain the isolate MIC). Each protocol has their own cut off values. The active CLSI protocols (M27 4th ed. M59 rd ed.) have epidemiological cut off values or ECVs (not breakpoints) for Cryptococcus neoformans VNI. For this genetic variant, fluconazole ECV is 8ug/ml. Thus, if the strain was not genotiped, and no susceptibility testing used method is mentioned no interpretation is posible.
Amended – a sentence commenting on CLSI ECVs as the current standard has been inserted and referenced. The susceptibility testing methodology has been amended as above. The methodology used for treatment in this case was an AUC:MIC ratio based on the model referenced in the text. This model was based on the MICs of 21,000 isolates from a systematic review, the genotype was not a consideration when this model was created.
Although established breakpoints for fluconazole in CM are MIC ≤8.0 mg/L suscep- 139
tible, 16-32 μg/mL dose-dependent susceptible and ≥64 μg/mL resistant, it is unclear 140
whether significantly adverse outcomes are seen in those with decreased susceptibility 14。
Reviewer 2 Report
This manuscript presented an interesting case, in which isavuconazole was successful utilized in the treatment of cryptococcal meningitis and cryptococcoma. Isavuconazole is a novel broad-spectrum triazole antifungal drug. Currently, it is mainly used in invasive infection caused by Aspergillus and Mucor. There were also a few reports on its utilization in treating cryptococcosis. Since the clinical strain isolated from CSF was resistant to fluconazole, isavuconazole was alternatively selected in maintenance treatment based on the drug sensitivity test. The clinical outcome was remarkable and encouraging. I think it’s of significance for clinical management of cryptococcosis patients infected with fluconazole-resistant strain. There are two suggestions.
- The patient's informed consent must be obtained before publication of the article. Please specify it in the manuscript.
- 2. The grammar needs to be further checked. Some punctuations were missing in the manuscript.
Author Response
Dear reviewer, thank you for taking the time to review this manuscript and your response. There is now a sentence outlining informed consent has been obtained after the Conclusion. The grammatical errors, we believe have been amended.